# Can Foliar Pulverization with CaCl_2_ and Ca(NO_3_)_2_ Trigger Ca Enrichment in *Solanum tuberosum* L. Tubers?

**DOI:** 10.3390/plants10020245

**Published:** 2021-01-27

**Authors:** Ana Rita F. Coelho, Fernando C. Lidon, Cláudia Campos Pessoa, Ana Coelho Marques, Inês Carmo Luís, João Caleiro, Manuela Simões, José Kullberg, Paulo Legoinha, Maria Brito, Mauro Guerra, Roberta G. Leitão, Carlos Galhano, Paula Scotti-Campos, José N. Semedo, Maria Manuela Silva, Isabel P. Pais, Maria J. Silva, Ana P. Rodrigues, Maria F. Pessoa, José C. Ramalho, Fernando H. Reboredo

**Affiliations:** 1Earth Sciences Department, Faculdade de Ciências e Tecnologia, Universidade NOVA de Lisboa, Campus da Caparica, 2829-516 Caparica, Portugal; fjl@fct.unl.pt (F.C.L.); c.pessoa@campus.fct.unl.pt (C.C.P.); amc.marques@campus.fct.unl.pt (A.C.M.); idc.rodrigues@campus.fct.unl.pt (I.C.L.); jc.caleiro@campus.fct.unl.pt (J.C.); mmsr@fct.unl.pt (M.S.); jck@fct.unl.pt (J.K.); pal@fct.unl.pt (P.L.); mgb@fct.unl.pt (M.B.); acag@fct.unl.pt (C.G.); mfgp@fct.unl.pt (M.F.P.); fhr@fct.unl.pt (F.H.R.); 2GeoBioTec, Faculdade de Ciências e Tecnologia, Universidade NOVA de Lisboa, Campus da Caparica, 2829-516 Caparica, Portugal; paula.scotti@iniav.pt (P.S.-C.); jose.semedo@iniav.pt (J.N.S.); abreusilva.manuela@gmail.com (M.M.S.); isabel.pais@iniav.pt (I.P.P.); mjsilva@isa.ulisboa.pt (M.J.S.); cochichor@isa.ulisboa.pt (J.C.R.); 3LIBPhys-UNL, Departamento de Física, Faculdade de Ciências e Tecnologia, Universidade NOVA de Lisboa, Campus da Caparica, 2829-516 Caparica, Portugal; mguerra@fct.unl.pt (M.G.); rg.leitao@fct.unl.pt (R.G.L.); 4INIAV-Instituto Nacional de Investigação Agrária e Veterinária, Avenida da República, Quinta do Marquês, 2780-157 Oeiras, Portugal; 5ESEAG-COFAC, Avenida do Campo Grande 376, 1749-024 Lisboa, Portugal; 6Plant Stress & Biodiversity Lab, Centro de Estudos Florestais (CEF), Instituto Superior Agronomia (ISA), Universidade de Lisboa (ULisboa), Quinta do Marquês, Av. República, 1349-017 Lisboa, Portugal; anadr@isa.ulisboa.pt

**Keywords:** calcium biofortification, photosynthesis, physical and organoleptic characteristics, tissue localization of calcium *Solanum tuberosum*

## Abstract

This study aimed to assess the efficiency of Ca enrichment in tubers of three genotypes of *Solanum tuberosum* L., through foliar spraying with CaCl_2_ and Ca(NO_3_)_2_ solutions. In this context, soil heterogeneity of three potato-growing fields, as well as the implications of Ca accumulation among tissues and some quality parameters were assessed. Three potato varieties (Agria, Picasso and Rossi) were grown in three production fields and during the life cycle, four pulverizations with calcium chloride (3 and 6 kg ha^−1^) or calcium nitrate (0.5, 2 and 4 kg ha^−1^) were applied. For screening the potential phytotoxicity, using Agria as a test system, the potential synthesis of photoassimilates was determined, and it was found that after the 3rd Ca application, leaf gas exchanges were moderately (net photosynthesis), to strongly (stomatal conductance) affected, although without impact on Ca accumulation in tubers. At harvest, the average Ca biofortification index varied between 5–40%, 40–35% and 4.3–13% in Agria, Picasso and Rossi, respectively. Moreover, the equatorial region of the tubers in general showed that Ca accumulation prevailed in the epidermis and, in some cases, in inner areas of the potato tubers. Biofortified tubers with Ca also showed some significant changes in total soluble solids and colorimetric parameters. It is concluded that Ca enrichment of potato tubers through foliar spraying complemented the xylem mass flow of Ca from roots, through phloem redistribution. Both fertilizers showed similar efficiency, but Rossi revealed a lower index of Ca accumulation, eventually due to different metabolic characteristics. Although affected by Ca enrichment, potato tubers maintained a high quality for industrial processing.

## 1. Introduction

Calcium is an essential nutrient, namely for bone development and functioning of the circulatory system of the human population [1,2,3,4]. Calcium deficiency in the human diet can also trigger pathologies (infantile bone deformity, osteoporosis, rickets and eventually hypertension and colorectal cancer) and high mortality in some world regions, namely Africa and Asia, where Ca intake is almost nil [2,5,6]. To surpass Ca deficiency, plants can provide the ultimate source of nutrients in the human diet, but most of the essential nutrients, such as Ca, are lacking in staple food [4,7]. Nonetheless, agronomic biofortification, a fast, reliable, sustainable, accessible, and cost-effective way to increase minerals content [8,9,10,11], can surpass Ca deficiency, allowing its enrichment in the edible parts of selected food crops. In fact, selection of basic crops for biofortification, consumed worldwide, can improve low-quality diets, where there is a limited choice of foods and the soils are devoid of bioavailable nutrients [10,12], affecting the uptake and translocation of minerals to the edible portions [7,13].

Calcium, although relatively immobile in cells and not readily remobilized from the mature to the active growing parts of plants [5], is an essential macronutrient for plant growth and development, being required as Ca^2±^. It is a transduction agent, providing stability and integrity to the cell wall [14]. It also plays a central role in stress responses [15] and is required, as a cofactor, by enzymes involved in the catabolism of ATP and phospholipids, further acting as a second messenger in metabolic regulation [16]. Nevertheless, gaps exist in our understanding of how Ca is transported within the plant [17]. Reports indicate that calcium fluxes from roots to the shoot largely occur from the root apex and/or regions of lateral root initiation, mostly implicating the root apoplasm (which is relatively non-selective between divalent ions), in regions where Casparian bands are absent or via the cytoplasm of unsuberisedendodermal cells where Casparian bands are present [18,19]. Thereafter, essentially in the apoplasm cell water space, Ca^2±^ binds to negatively charged residues and are taken up by cells down the electrochemical gradient for Ca^2±^, or pass through the water-free space of the cell wall to the xylem [17]. Once in the xylem, through a Ca^2±^ or complexed with organic acids transport pathway [20], translocation occurs via mass flow with the transpiration stream [21]. This link between transpiration and Ca transport further highlights the low rate of symplasmic transport of Ca^2±^ (i.e., through the cytoplasm of cells linked by plasmodesmata). However, it was alternatively reported that portioning of xylem Ca to adaxial and abaxial epidermal cells occurs and that the contents of this nutrient in the xylem sap does not comply with the mass flow [22,23]. On the other hand, several authors reported that Ca is phloem immobile and does not redistribute in plants [17,24,25,26], but the opposite was also presented by other authors. For instance, Nelson et al. [27] localized radioactive strontium (a calcium analog) in the phloem tissue of potato tubers, yet they could not conclude if strontium reached the sieve tubes. Davies and Millard [28] also found ^45^Ca in the phloem tissue in potato tubers, and used the Ca/sucrose ratio in the sap to conclude that Ca kinetics in the phloem may account for a significant proportion of the potato tuber calcium. Oparka and Davies [29] using microscopic techniques further revealed that Ca occurs in companion cells and phloem parenchyma.

Tuberization in potato plants is controlled by environmental and nutritional factors, which affect the levels of endogeneous growth substances [30,31]. At a nutritional level the intracellular Ca, rather than Ca in the free space, plays an important role in the tuberization process [30], namely through a signaling pathway implicating Ca/calmodulin [32]. However, it seems that addition of Ca to the soil (namely, CaCl_2_ and Ca(NO_3_)_2_) during tuberization can reduce tubers number, suppressing the tuberization signal by increasing gibberellin content [33]. To avoid this effect, supplemental foliar application of CaCl_2_ and Ca(NO_3_)_2_ is suggested by the PEI (Nutrient Management–Prince Edward Island) [34] and carried out at a Ca rate of 0.5–1 kg ha^−1^ (two to four applications, two weeks apart to avoid deficiencies, with the first application occurring at full bloom). Under this workflow, El-Hadidi et al. [35] found that, through foliar pulverization, the Ca levels significantly increased in potato leaves and tubers. El-Zohiri and Asfour [36], through pulverization with Ca(NO_3_)_2_ also found an increased content of Ca in potato tubers and Seifu and Deneke [37], spraying with CaCl_2_ and Ca(NO_3_)_2_ also obtained a similar trend.

As *Solanum tuberosum* L. is the 4th most cultivated staple food crop worldwide (after wheat, rice and maize), this study aimed to assess the efficiency of Ca enrichment in tubers of three species, through foliar spraying with CaCl_2_ and Ca(NO_3_)_2_. In this framework soil heterogeneity of three potato-growing fields, as well as the implications of Ca accumulation among tissues, and some quality parameters, were further considered. Potato varieties Agria, Picasso and Rossi were chosen, as test systems, due to its industrial relevance. They have high yields, with Agria further being suitable for a long frying time without browning and dehydrated products, whereas Picasso presents an excellent culinary quality, not discoloring after cooking and the Rossi variety is full of flavor.

## 2. Results

Geomorphology (Figure 1; Table 1) of potato-growing fields strongly affects water surface drainage. After slopes calculation of each plot of land (Table 1; Figure 1B,C), drainage, or surface drainage zones, from the planar regions that accumulates surface water (and, consequently, promote its infiltration into the ground) were differentiated into classes in maps. It was found that the Moledo field had a greater aptitude for accumulation of surface water, and/or infiltration (with approximately 54% of the area with flat or smooth slope morphology), while Casal Galharda only presented 21% of the area with low drainage capabilities (Table 1). Approximately 99.5% of the area of the Boas Águas field (Table 1) had moderate drainage capabilities, thus with poor conditions for accumulation and/or water infiltration (therefore promoting runoff). Moreover, as these slope maps were obtained before crop implementation (Figure 1D), they allowed the assessment of the initial land state of the field. In this context, the NDVI (Normalized Difference Vegetation Index) index indicated, as expected, that soils of the three experimental fields were dominantly without vegetation (red and yellow colors), except for the green areas, where the presented vegetation did not match the crops in this study (Figure 1A,D).

The organic matter and the electrical conductivity in the soils of Moledo were significantly lower relatively to the soils of Casal Galharda and Boas Águas (Table 1). The electrical conductivity of the soil from Boas Águas and Casal Galharda was 60–70% higher than in Moledo (Table 1), indicating a higher salt content (thus, implicating greater energy expenditure for water absorption due to the osmotic effect) by roots. In the three experimental fields pH did not vary significantly, ranging from 7.30 to 7.41 (as such, soils are slightly basic). Moledo also showed the lowest contents of Ca, K, Mg, S, Zn and Mn, whereas Boas Águas exhibited the highest levels of Ca, K and Mg and Casal Galharda displayed the highest concentrations of P, Fe, S, Zn, and Mn (Table 1).

The irrigation water in the three potato fields was of underground origin, with predominance of bicarbonate and calcium sulfate (Moledo and Boas Águas) and calcium bicarbonate (Casal Galharda), with high salinity (concentration of salts evaluated, in terms of electrical conductivity, between 750 and 2250 μS.cm^−1^, at 20 °C). These irrigation water belong to class C3S1, with SAR (sodium adsorption ratio) index 0.64 (Moledo), 0.75 (Boas Águas) and 1.19 (Casal Galharda). Additionally, these waters were sub-saturated with calcium carbonate, having pH of 7.3, 7.4 and 7.4, and an ISL (Langelier saturation index) of −0.41, −0.5 and −0.18, in Moledo, Casal Galharda and Boas Águas fields, respectively.

As the synthesis of photoassimilates would be impaired if the threshold of toxicity is reached, using Agria variety as a potato test system, after the 3rd foliar spraying several leaf gas exchange parameters (net photosynthesis rate: P_n_; stomatal conductance to H_2_O vapor; g_s_; internal [CO_2_]: C_i_; transpiration rate:E; insytantaneous water use efficiency: iWUE were monitored to assess potential stresses (Figure 2). It was observed that, relatively to the control, P_n_ showed lower values in all treatments (between 15 and 21%), significantly in some treatments with CaCl_2_ (3 kg ha^−1^) and Ca(NO_3_)_2_ (0.5 kg ha^−1^). The g_s_ was an even more marked reduction than P_n_, with significant decreases for all treatments and products between 40–53%, whereas C_i_ also significantly reduced between 28–42%. In contrast, iWUE increased in most treatments, reaching ca. 11% higher values with Ca(NO_3_)_2_ treatments and between 22–38% with CaCl_2_.

Calcium accumulation in the whole tubers was assessed, at harvest, in the three fields. Relative to the control, the content of Ca in Agria was (Table 2) significantly higher in all treatments (except with Ca(NO_3_)_2_–0.5 kg ha^−1^), with biofortification indexes ranging between 5% and 40% (maximum levels with CaCl_2_–6 kg ha^−1^ and Ca(NO_3_)_2_–4 kg ha^−1^). Also relative to the control, the levels of Ca in Picasso varied among treatments (with Ca(NO_3_)_2_ or CaCl_2_), with the higher concentrations being observed in the highest treatments with CaCl_2_ and Ca(NO_3_)_2_ (biofortification indexes of 40–35%). Calcium biofortification indexes in Rossi, relative to the control, ranged (Table 2) between 4.3–13% (maximum levels with Ca(NO_3_)_2_–4 kg ha^−1^).

Independent of the biofortification index, when five transverse sections of the equatorial region in the three varieties of potato tubers were considered, the highest contents of Ca were among those close to the epidermis (Table 3; Figure 3). Moreover, relative to the control, treatments with CaCl_2_ also revealed higher contents in all the transverse sections of the equatorial region of tubers of Agria and Rossi varieties. Additionally, in Picasso tubers, Section 2, Section 4 and Section 5 showed lower values relatively to the control. Calcium nitrate treated tubers of Agria and Picasso also showed higher contents of Ca than the control (in all the transverse sections of the equatorial zone), but variety Rossi showed an heterogeneous pattern (with higher values relatively in the control only in Section 1 and Section 5).

In Agria, relative to the control, dry weight decreased significantly in Ca(NO_3_)_2_ treatments (Table 4). Relatively to the control, Picasso tubers showed a significant decrease only in treatment 0.5 kg ha^−1^ Ca(NO_3_)_2_, but higher values were also obtained with 4 kg ha^−1^ Ca(NO_3_)_2_ and 6 kg ha^−1^ CaCl_2_ (Table 4). The highest values in the tubers of Rossi were found in the control (Table 4).

Relative to the control, Agria tubers showed significantly lower content of total soluble solids in the 4 kg ha^−1^ Ca(NO_3_)_2_ treatment, whereas Picasso tubers only revealed significantly higher values in treatment with 3 kg ha^−1^ CaCl_2_ (Table 4). Rossi tubers showed significantly lower values in 0.5 and 2 kg ha^−1^ Ca(NO_3_)_2_.

At harvest, scanning colorimetric analysis in the visible spectral region (450–650 nm) of the three pulp tubers revealed a maximum transmittance at 550 nm (Figure 4), corresponding to yellow. Considering that color is a quality parameter in food, if was found that there was not any relevant change in the pulp color of the tubers regarding the Ca applications.

## 3. Discussion

Mineral accumulation in plants is closely linked to soil composition, electrical conductivity, content of organic matter, runoff water and quality and weather conditions. However, mineral enrichment in plants further requires a specific workflow and selection of varieties prone to accumulation. In this context, the edaphoclimatic characteristics of the three potato-growing fields (Table 1; Figure 1) and their water quality were considered for Ca natural fortification of the three potato varieties of *S. tuberosum*. To determine the efficiency of Ca enrichment of potato tubers through foliar spraying, as well as the accumulation implications among tissues, Moledo field was chosen as a contrasting test system (due to soil and water irrigation characteristics), relative to the fields Casal Galhardas and Boas Águas. In this context, the experimental areas in each potato fields were selected (Figure 1) to minimize the effects of runoff water [38]. As soil is the primary source for potato plants to receive mineral elements through their roots [7,12], in all these fields the slightly basic pH of the soils was considered, since its acidification is frequently associated with deficiency of essential plant cations like Ca in potato crops [39]. Also, as Ca is the third most available nutrient in soils [40], being an essential nutrient for plants in the form of Ca^2+^ [15], the positive correlation between exchangeable Ca and soil organic carbon in the soil (i.e., an increase in soil organic carbon concentration generally increases the cation exchange capacity) must be considered [41] for potato production. Accordingly, the lowest levels of Ca and organic matter of the experimental field Moledo was further used to assess potential interferences relative to the remaining fields (Table 1). The lower levels of K, Mg, S, Zn and Mn in the soil of the Moledo field was also considered. Indeed, Ca^2+^ is a dominant cation in slightly alkaline reaction soils, organic matter has lower levels of colloidal materials and subsists a cationic antagonism between Ca and Mg or K (i.e., high levels of one or more of these nutrients can result in decreased uptake of another, despite soil levels) [42]. On the other hand, in the Moledo field, the low levels of S, Zn and Mn additionally further limited the synergistic interaction of Ca with these nutrients [43,44]. Considering the irrigation water of the Moledo field, the Langelier saturation index also revealed a slightly higher tendency to dissolve calcium carbonate (Table 1). Nevertheless, there was no danger of increased soil alkalization, in all the potato-growing fields, due to the low Na^+^ concentration relative to Ca^2+^ and Mg^2+^. Indeed, as potato crops are moderately sensitive to salinity [45,46], lowering potato plant emergence, enhancing haulm senescence and reducing the growth of both haulm and tubers [47], a long-term potential interference with the Ca biofortification workflow was minimized. Additionally, the irrigation water of the Moledo field further consolidated its use as a contrasting test system, relative to the other potato fields, since this revealed low accumulation of K^+^, Mg^2+^, Cl^−^, HCO_3_, SO_4_^2−^ and PO_4_^3−^. Moreover, high electric conductivity (as shown among the potato fields), although closely linked with high levels of salts, did not reached critical levels to determine potato seed tubers, delaying root and shoot development and shoot emergence, which is likely caused by inhibition of the cell division and elongation of the sprout meristems [45].

Calcium is implicated in a wide number of physiological processes, including growth and development as well as tolerance to environmental stresses. The background of these interacting effects is the photosynthetic functioning, yet depending on Ca contents, negative impacts can develop [48,49]. In fact, Ca^2+^ is involved in the regulation of the photosynthetic pathway (namely, stomatal closure, to photosystems functioning, non-photochemical quenching and xanthophyll cycle) [49,50]. As the synthesis of photoassimilates is impaired if the threshold of toxicity is reached, using the variety Agria as a test system, it was found that, regardless of the applied dose or product, Ca-biofortification promoted a low-to-moderate decrease in P_n_, and an even stronger decline of g_s_ with a consequent iWUE increase, (Figure 2). Although P_n_ reduction may be associated to reductions in maximum and current efficiency of photosystem II and in the quantum yield of the photosynthetic electron transport, the strong stomatal closure led to marked declines in the internal [CO_2_] (C_i_), thus pointing to a stomatal limitation on P_n_. In fact, stomata guard cells integrate a response to environmental and endogenous signals to regulate stomatal opening and closure, with cytoplasm Ca^2+^ oscillations playing a crucial role in this process [51]. Furthermore, external Ca concentration were observed to promote stomatal closure, likely through interference on Ca^2+^-sensing receptor (CAS) protein [51,52], which seems to be the case, as regards g_s_ reduction in Agria plants. Still, these impacts at the photosynthetic performance had low, if any, impact in the Ca enrichment performance, as its accumulation and tubers weight were not inhibited (Table 2, Table 3 and Table 4).

After tuberization, Ca accumulation in tubers occurred thought foliar spraying with CaCl_2_ or Ca(NO_3_)_2_, still without visible toxicity symptoms. Nonetheless, the same accumulation indexes in Agria and Picasso potatoes were obtained (5–40% considering all treatments) in spite of the contrasting soil and water irrigation properties (Table 1, Table 2 and Table 3) prevailing in the relative production fields. Furthermore, it was interesting was to notice that in spite of similar soil and irrigation water properties in Casal Galharda and Boas Águas fields, the different potato genotypes (Picasso and Rossi, respectively) revealed quite different Ca accumulation indexes (5–40% and 4.3–13%, respectively) (Table 2). Accordingly, these data clearly pointed that CaCl_2_ and Ca(NO_3_)_2_ spraying surpassed the limits imposed by soil and water irrigation characteristics (with similar efficiency), whereas Rossi metabolism seems to be less prone to Ca accumulation in the tubers. Accordingly, independently of the accumulation extend in the potato tubers in the different genotypes, our data suggested a Ca^2+^ mass flow through the xylem from roots and coupled to the transpiration stream [7,17,21,24,53] complemented, as suggested by several authors [27,28,29], with phloem redistribution of Ca provided through foliar spraying. Indeed, as previously suggested [34,35,36,37] our data strongly indicated that foliar spraying with CaCl_2_ and Ca(NO_3_)_2_ can enrich potato tubers with Ca. Besides, as in the three varieties Ca content prevailed close to the epidermis, as also seen by [7] working with other variety (‘Stirling’), our data further showed that Ca translocation from the shoot did not affect the pattern of this nutrient accumulation in the tissues of tubers (Table 3; Figure 3).

Some studies (namely with apples, mangos, potatoes, kiwifruits, peaches and strawberries) showed some beneficial effects on the physiological and quality parameters by supplying Ca, using CaCl_2_ or Ca(NO_3_)_2_ [5]. In this context, dry matter content is one of the characteristics that most determines the texture of the tubers, being a criterion to the classification of potato tubers quality [54]. When higher than 20%, is also compatible with the requirements of the processing industry. As such, tubers with high dry matter content display a greater industrial performance, which was our case upon Ca enrichment. Indeed, upon Ca enrichment, tubers of the three varieties only showed some minor deviations on dry weight and total soluble solids contents (Table 4), which indicated that foliar spraying of the three genotypes also seems to have beneficial effects on yield. Indeed, considering that the intracellular Ca interferes in the tuberization process, through a signaling pathway implicating Ca/calmodulin [30,32], foliar spraying further seemed to alleviate the suppression tuberization signal triggered by increasing gibberellin content when CaCl_2_ and Ca(NO_3_)_2_ are applied in soils [33].

Considering that total soluble solids is another quality parameter, as Ca enriched tubers of the three genotypes, relatively to the control, displayed significant differences among treatments (Table 4), spraying to some extend affected negatively this parameter in Agria and Rossi. Nevertheless, it must be pointed out that all the potato genotypes showed higher values relative to other potato cultivars [55], such as Ágata (4.00), Atlantic (4.8) and BRS Clara (3.2). Higher values were also found when comparing Agria, Picasso and Rossi with Asterix (4.25) and Mondial (3.95) [56]. The interference of Ca biofortification on color parameters was negligible, further indicating the absence of depreciative effects.

## 4. Materials and Methods

Three experimental potato-growing field (Moledo, Casal Galharda and Boas Águas), located in Western Portugal (GPS coordinates according to European Datum-39°16′38.77″ N; 9°15′8.294″ W, 39°16′12.576″ N; 9°14′14.492″ W and 38°16′31.76″ N; 9°13′46.77″ W), were used to grow three potato (*Solanum tuberosum* L.) genotypes (Agria, Picasso and Rossi, respectively). During the agricultural period, from 4 May to 25 September 2018, air temperatures reached an average daily of 23 °C and 15 °C (with maximum and minimum values of 41 °C and 6 °C, respectively). The average rainfall was 0.41 mm, with a daily maximum of 18.03 mm and accumulation of 60.4 mm (Figure 5).

Orthophotomaps, of the experimental fields of the growing potato varieties, were produced using a high-definition and multi-sector RGB camera (with three electromagnetic spectra bands—red, green and blue) and a parrot sequoia camera (with five electromagnetic spectra bands—NIR (near infrared), REG green, red and RGB) installed in a drone. The calibration of the camera (parrot sequoia), took into account the environmental brightness conditions. The images were processed with Workstation (AORUS, GIGA-BYTE Technology Co., Ltd. 2019, Croatia). The drainage patterns of surface water and the geomorphology of the field was studied with an Agisoft PhotoScan Professional (Version 1.2.6, Software of 2016 and the ESRI of 2011 and ArcGIS Desktop-Release 10 from Redlands, CA: Environmental Systems Research Institute). The classification of, surface water drainage areas followed [57]. The lower class corresponded to flattened surfaces, because of the accumulation of surface water, representing potential infiltration areas. The highest class represented zones that, due to its morphology, promote the surface water runoff, having a reduced amount of water infiltration.

Soils of the experimental fields adopted the usual set of actions in potato cultivation. Before planting, organic fertilizer (organic fertilizer folhadoro 4–3–3) and Nergetic 10–10–22 were applied in the soils. The organic fertilizer folhadouro 4–3–3), is a 100% organic pellet fertilizer, ecological and hygienic, containing mainly humic acids, that facilitate the availability of nutrients, gradually releasing the minerals present in the soil. The Nergetic fertilizer 10–10–22, was applied in the three fields as a surface fertilizer, because has a macromolecule that regulates leaching and volatilization. It is a complex NPK (Nitrogen, Phosphor and Potassium), fertilizer with Mg and S.

In the experimental fields of the growing potato varieties, a hexagonal grid (4.50 m × 6.60 m) was applied and 16 soil samples (100 g, picked up at 30 cm depth) were collected for physical and chemical analysis. Soil samples were passed through a 2 mm nylon sieve to remove major debris before analysis. After drying at 105 °C, for 24 h, until constant weight, soil moisture was determined. Organic matter (OM) was then estimated after combustion for 4 h at 550 °C. Following [58], quantification of mineral elements was carried out by X-ray fluorescence, using a Thermo Scientific Niton XL3t 950 He GOLDD ± XRF. Electrical conductivity (EC) and pH were measured with a multiparameter analyzer (C 6030) and SP21 (pH) and SK20T (CE) electrodes, in a soil mixture with water (1:2.5 g _soil_ mL^−1^
_water milli-q_) under stirring after a thermal bath (25 °C) for 30 min [59].

Water quality of the fields considered physical (pH, temperature and electrical conductivity) and chemical (bicarbonate, sulfate, chloride, sodium, calcium, magnesium, potassium, nitrate and phosphate) parameters. Electrical conductivity (EC) and pH were determined using a Consort multiparameter analyzer (C 6030) and SP21 (pH) and SK20T (CE) electrodes. Calcium, Na, K and Mg ions were quantified using a Metrohm (Model 761 Compact IC) chromatograph, equipped with column and pre-column (Metrosep cation 1–2, 6.1010.000), using an eluent mixture (4 mM tartaric acid/1 mM dipicolinic acid) at a flow rate of 1.00 mL/minute and a sample injection of 10.0 μL. Alkalinity/bicarbonate was determined by titration, in 100 mL of water samples, using 0.1 N hydrochloric acid as titrant, in the presence of 0.1% methyl orange [60]. Chloride, sulphate, nitrate and phosphate ions were quantified by photometry, using specific kits from Spectroquant NOVA 60, Merck (1.14897, 1.14779, 1.14773 and 1.14842). Water classification in the soils of the three fields, considering dominant ions, followed [61]. Sodium adsorption index was determined and related to the electrical conductivity, in classes C and S. The Langelier saturation index was also estimated (at 20 °C) from the pHe (equilibrium pH), to determine the fouling or aggressiveness of the water relatively to calcium carbonate.

Potato genotypes Agria, Rossi and Picasso were sown in 4, 11 and 15 of May. After the beginning of tuberization, four foliar spraying (with 8–10 days interval) with CaCl_2_ (3 and 6 kg ha^−1^) or Ca(NO_3_)_2_ (0.5, 2 and 4 kg ha^−1^) solutions were performed (Figure 5). Control plants were not sprayed at any time with Ca(NO_3_)_2_ or CaCl_2_. In plots 20 × 20 m (Moledo and Casal Galharda) and 20 × 24 m (in Campo Boas Águas), each plot contained the six treatments (control included), having been carried out in quadruplicate (compass, 60–80 cm). Tubers from Moledo, Casal Galharda and Boas Águas fields were harvested on 4/9/2018, 24/9/2018 and 25/9/2018, respectively.

Leaf gas exchange parameters were determined, using as a test system the Agria genotype, according to [62], in 6 randomized plants per treatment. Net photosynthesis rate (P_n_), stomatal conductance to water vapor (g_s_), CO_2_ internal concentration (C_i_), and transpiration rate (E) were measured under photosynthetic steady-state conditions after at least 2 h of illumination (usually between 10:30 and 11:30). A portable open-system infrared gas analyzer (Li-Cor 6400, LiCor, Lincoln, NE, USA) was used under environmental conditions, with a photosynthetic photon flux density (PPFD) ranging between 1200–1400 µmol m^−2^ s^−1^, air temperature of 33.8 ± 0.4 °C, and external CO_2_ (ca. 400 ppm). The ratio of P_n_-to-E (representing the units of assimilated CO_2_ per unit of water lost through transpiration) allowed the calculation of leaf instantaneous water-use efficiency (iWUE).

After harvest in the experimental potato-producing fields, eight randomized tubers (with similar size) were washed, dried at 60 ºC until constant weight and grounded in an agate mortar. After that, the homogenate was divided into four samples (*n* = 4) and an acid digestion procedure was performed with a mixture of HNO_3_- HClO_4_ (4:1) according to [63,64]. After filtration, Ca content was measured by atomic absorption spectrophotometry, using a model Perkin Elmer AAnalyst 200, and the absorbency in mg/L was determined with a coupled AA WinLab software.

Location of Ca in the tissues of the tubers collected at harvest was determined using the µ-EDXRF system (M4 Tornado™, Bruker, Germany), as previously described in detail for food matrixes [65,66]. The X-ray generator was operated at 50 kV and 100 µA without the use of filters, to enhance the ionization of low-Z elements. For a better quantification of Ca, a set of filters between the X-ray tube and the sample, composed of three foils of Al/Ti/Cu (with a thickness of 100/50/25 µm, respectively) was used. All the measurements with filters were performed with 600 µA current. Detection of fluorescence radiation was performed by an energy-dispersive silicon drift detector, XFlash™, with 30 mm^2^ sensitive area and energy resolution of 142 eV for Mn Kα. To better measure the distribution mapping of Ca, the tubers were cut, at the equatorial region, into slices with a stainless-steel surgical blade. Measurements were carried out under 20 mbar vacuum conditions. These point spectra were acquired during 200 s. The values of the content of Ca were obtained through the average of four readings taken by the device.

Dry weight was performed considering four randomized tubers per treatment (*n* = 4). Total soluble solids (an indication of sucrose, glucose, and fructose) was also measured in the juice of four randomized tubers per treatment (*n* = 4), using a digital refractometer Atago (Atago, Tokyo, Japan). Colorimetric parameters were determined in the pulp of four fresh tubers per treatment (*n* = 4) with a scanning spectrophotometric colorimeter (Agrosta, European Union). The sensor provides a 40 nm full-width half-max detection, covering the visible region of the electromagnetic spectrum. This sensor has 6 phototransistors with sensibility in a specific region of the spectrum (380 nm—violet; 450 nm—blue; 500 nm—green; 570 nm—yellow; 600 nm—orange; 670 nm—red). Light was furnished by a white light-emitting diode (LED) covering all the visible region.

Data were statistically analyzed using a one-way analysis of variance (ANOVA) to assess differences among treatments within potato cultivars, followed by a Tukey’s for mean comparison. A 95% confidence level was adopted for all tests.

## 5. Conclusions

Despite some minor heterogeneity of the edaphic characteristics of the potato-growing experimental fields and irrigation water, the tuber varieties Agria, Picasso and Rossi, did not show relevant symptoms of toxicity (as seen by the rates of photoassimilates synthesis) upon foliar spraying with CaCl_2_ or Ca(NO_3_)_2_. Under the technological workflow applied for Ca enrichment of potato tubers, foliar spraying complemented the xylem mass flow of Ca from roots coupled to shoot transpiration, through phloem redistribution. Both fertilizers showed similar efficiency, but Rossi revealed a lower index of Ca accumulation, eventually due to specific metabolic characteristics. Nevertheless, Ca deposition in tubers tissues prevailed close to the epidermis, but triggered some minor changes in the dry matter, total soluble solids and colorimetric parameters. From an agronomic perspective, our data indicate that foliar spraying with Ca can additionally favor this nutrient accumulation in potato tubers.

## Figures and Tables

**Figure 1 plants-10-00245-f001:**
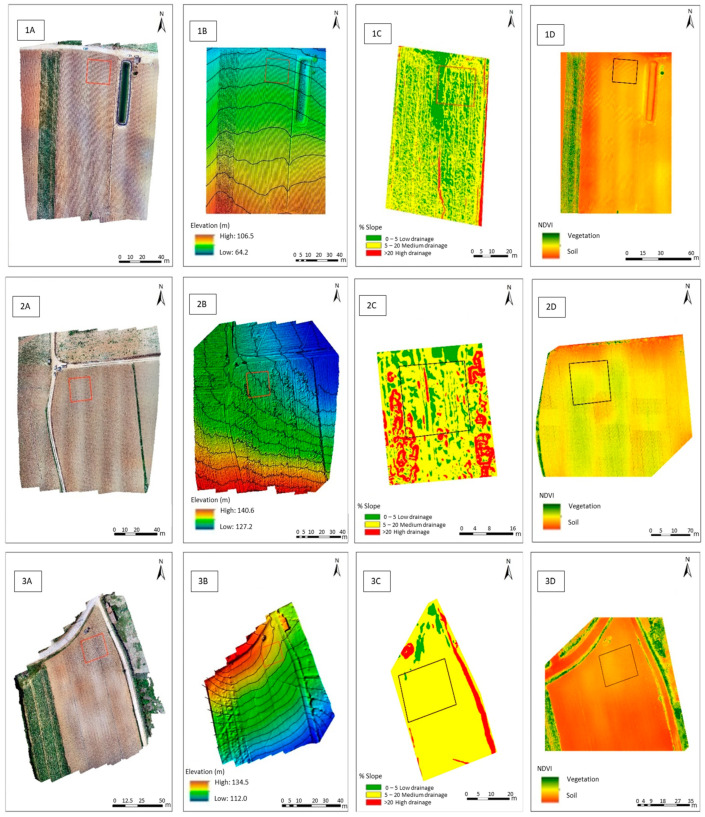
Orthophotomaps of *S. tuberosum* fields (1-Moledo, 2-Casal Galharda and 3-Boas Águas). Indication (in red) of limits of the three fields (**1A**,**2A**,**3A**); Digital elevation model of the fields (**1B**,**2B**,**3B**); Digital map of slopes of the fields (**1C**,**2C**,**3C**); NDVI model of the fields (**1D**,**2D**,**3D**). Information collected before tubers harvesting and biofortification treatments (18 May for the three fields).

**Figure 2 plants-10-00245-f002:**
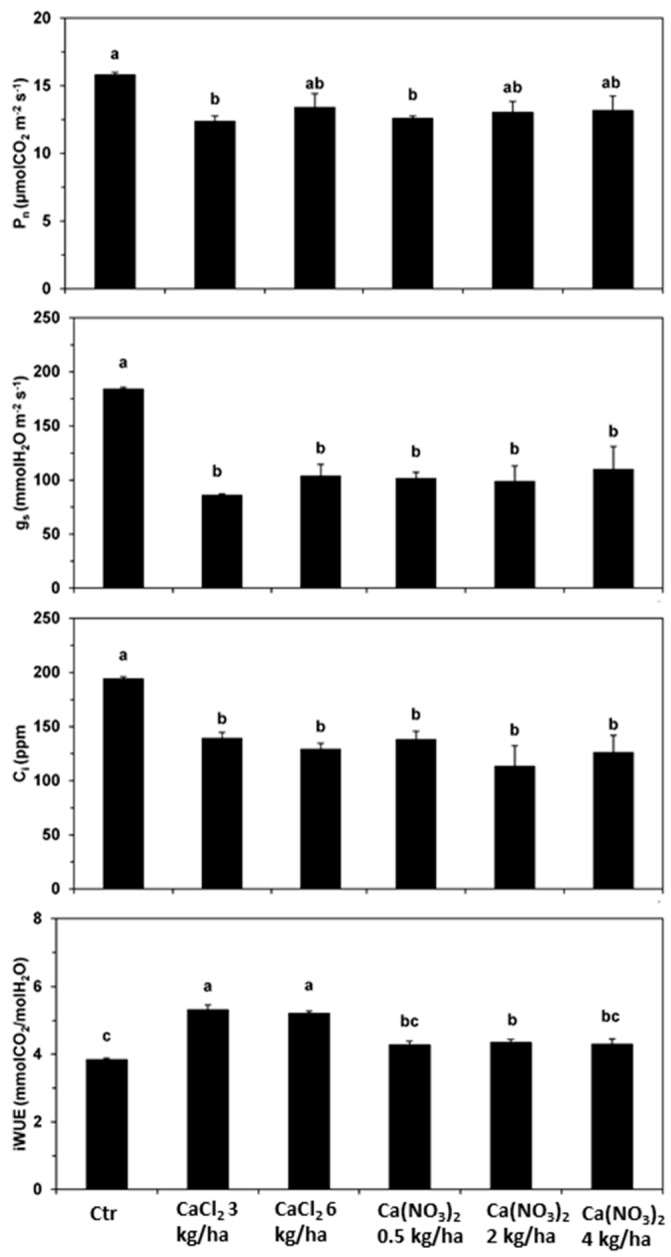
Leaf gas exchange parameters, net photosynthesis rate (P_n_), stomatal conductance to water vapor (g_s_), internal CO_2_ concentration (C_i_), as well as variation in the instantaneous water use efficiency (iWUE = P_n_/E) in leaves of *S. tuberosum* Agria variety, at 2nd August 2018 (after the 3rd leaf application). The concentrations of P_n_, g_s_ and iWUE ± S.E. (*n* = 6); Letters a, b and c indicate significant differences, of each parameter, between treatments (statistical analysis using the single factor analysis of variance (ANOVA) test, *p* ≤ 0.05).

**Figure 3 plants-10-00245-f003:**
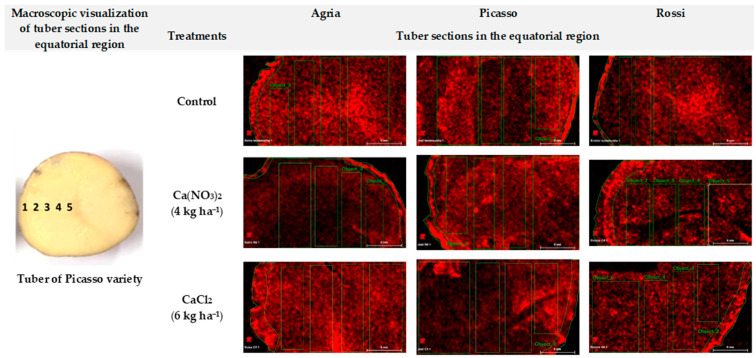
Tuber sections (from the epidermis to the center, 1–5) of each treatment, from the equatorial region of the tubers of *S. tuberosum* varieties (Agria, Picasso and Rossi), at harvest.

**Figure 4 plants-10-00245-f004:**
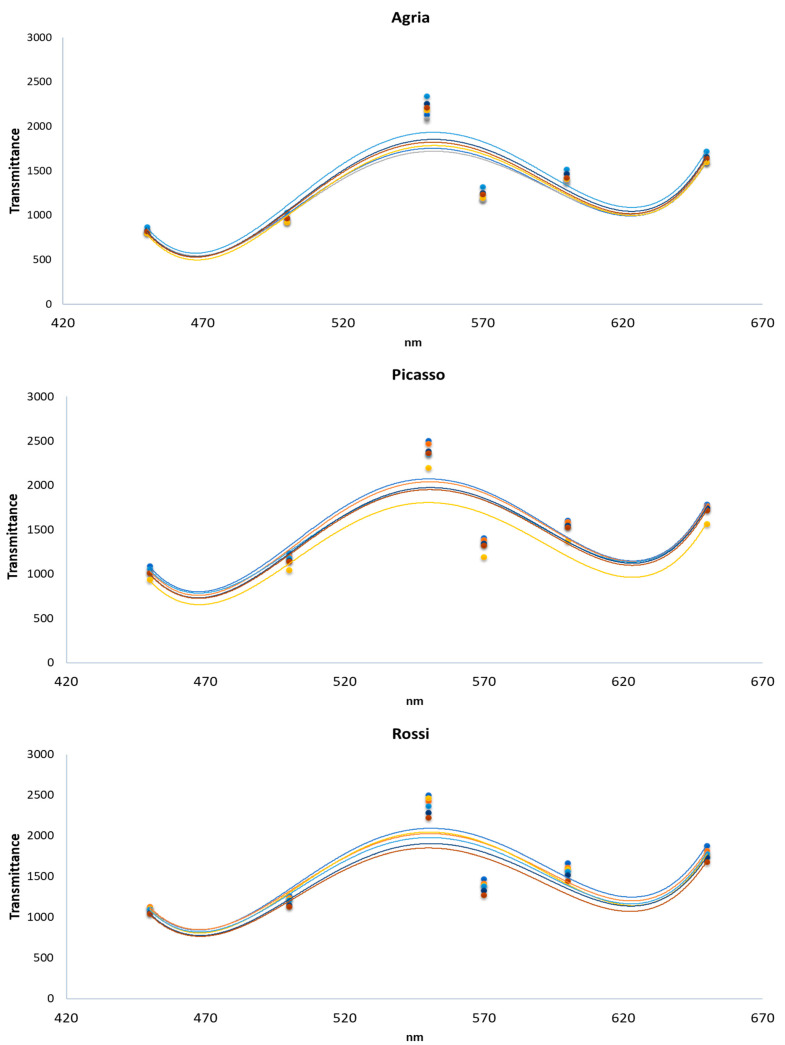
Visible spectra showing the average of transmittance (*n* = 3) in tubers of *S. tuberosum* varieties (Agria, Picasso and Rossi), at harvest ● Control, ● 0.5 kg ha^−1^ Ca(NO_3_)_2_, ● 2 kg ha^−1^ Ca(NO_3_)_2_, ● 4 kg ha^−1^ Ca(NO_3_)_2_,● 3 kg ha^−1^ CaCl_2_ and ● 6 kg ha^−1^ CaCl_2_).

**Figure 5 plants-10-00245-f005:**
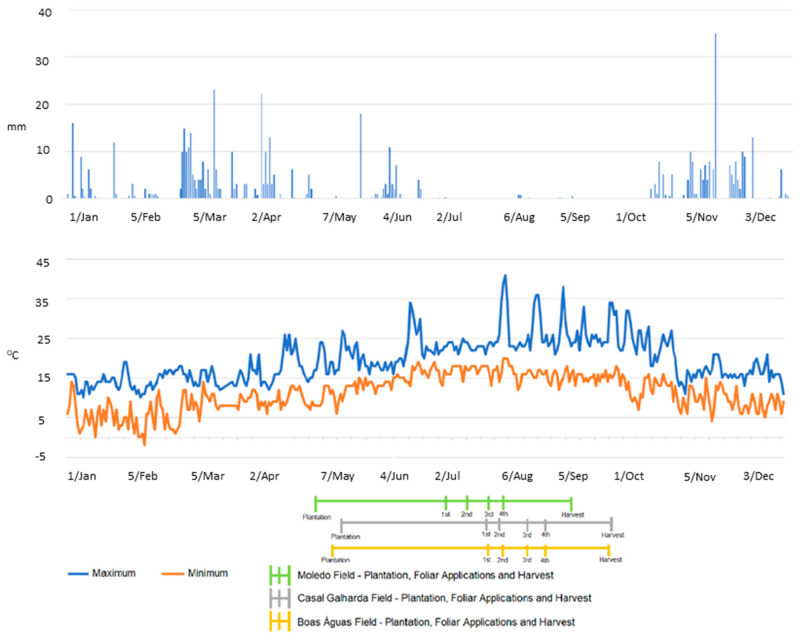
Humidity (mm) and maximum and minimum temperatures (°C). Indication of plantation, foliar applications, and harvest in the S. *tuberosum* fields (Moledo, Casal Galharda and Boas Águas).

**Table 1 plants-10-00245-t001:** Physical and chemical parameters of the soil and irrigation water of the experimental potato-growing fields Moledo, Casal Galharda and Boas Águas) selected for Ca biofortification of Agria, Picasso and Rossi varieties, respectively.

Field	Ability to Accumulate or Drain Surface Water										
	Slope Class (%)	Surface Drainage	Area (m^2^)					Área (%)			
**Moledo**	1-[0–5%]	Low	238.0					53.7			
	2-[5–20%]	Moderate	212.9					47.2			
	3- >20%	High	0.5					0.1			
**Casal Galharda**	1-[0–5%]	Low	76.8					21.1			
	2-[5–20%]	Moderate	265.5					72.9			
	3- >20%	High	21.7					6.0			
**Boas Águas**	1-[0–5%]	Low	2.5					0.5			
	2-[5–20%]	Moderate	475.4					99.5			
	3- >20%	High	0.1					-			
	**Soil analysis (0–30 cm deep) (*n* = 16)**										
	pH	Electrical Conductivity	Organic Matter	Ca	K	Mg	P	Fe	S	Zn	Mn
		μS cm ^−1^	%						ppm		
**Moledo**	7.41 ± 0.03 a	205 ± 17.5 b	1.88 ± 0.12 c	0.39 ± 0.03 b	2.20 ± 0.03 b	0.15 ± 0.01 b	0.23 ± 0.01 b	1.19 ± 0.07 b	55.9 ± 4.80 b	19.6 ± 1.54 c	318 ± 26.5 b
**Casal Galharda**	7.40 ± 0.05 a	349 ± 24.2 a	4.59 ± 0.10 a	0.65 ± 0.03 a	2.23 ± 0.07 b	0.19 ± 0.02 b	0.29 ± 0.01 a	2.59 ± 0.06 a	77.6 ± 1.11 a	62.7 ± 2.77 a	703 ± 56.8 a
**Boas Águas**	7.30 ± 0.06 a	332 ± 13.7 a	4.13 ± 0.14 b	0.71 ± 0.07 a	2.64 ± 0.02 a	0.24 ± 0.01 a	0.19 ± 0.01 c	0.50 ± 0.07 c	66.6 ± 1.24 b	41.7 ± 1.62 b	270 ± 31.1 b
	**Water analysis**										
	pH	Electrical Conductivity	Ca^2±^	K^±^	Mg^2±^	Na^±^	Cl^−^	HCO_3_^−^	SO_4_^2−^	NO_3_^−^	PO_4_^3−^
		μS cm ^−1^	mg L^−1^ (meq L^−1^)								
**Moledo**	7.2	1322	156.4(7.8)	2.1 (0.05)	22.5 (1.8)	22.8 (1.4)	56.6 (1.6)	297.6 (4.8)	166 (3.4)	90.3 (1.4)	<1.5 (<0.04)
**Casal Galharda**	6.9	1340	119.2(5.9)	4.6 (0.1)	37.1 (3.0)	56.3 (2.4)	89 (2.5)	374.5 (6.1)	164 (3.4)	1.2 (0.01)	<1.5 (<0.04)
**Boas Águas**	6.9	1381	169.6(8.4)	2.2 (0.06)	20.6 (1.7)	41.2 (1.7)	69 (1.9)	330.6(5.4)	234 (4.8)	26.7 (0.4)	<1.5 (<0.04)

Means in the same column, not followed by a common letter, are significantly different (*p* < 0.05).

**Table 2 plants-10-00245-t002:** Mean values of Ca contents ± S.E. (*n* = 4), at harvest, in whole the tubers of *Solanum tuberosum* L., varieties Agria, Picasso and Rossi.

Treatments		Agria	Picasso	Rossi
		Ca (%)		
**Control**		0.020 ± 0.001 b	0.020 ± 0.001 c	0.023 ± 0.000 a
**Ca(NO_3_)_2_**	**0.5 kg ha^−1^**	0.021 ± 0.000 b	0.021 ± 0.002 bc	0.024 ± 0.001 ab
	**2 kg ha^−1^**	0.027 ± 0.000 a	0.026 ± 0.001 ab	0.025 ± 0.000 ab
	**4 kg ha^−1^**	0.028 ± 0.001 a	0.027 ± 0.001 ab	0.026 ± 0.000 b
**CaCl_2_**	**3 kg ha^−1^**	0.027 ± 0.002 a	0.023 ± 0.001 abc	0.023 ± 0.001 a
	**6 kg ha^−1^**	0.028 ± 0.000 a	0.028 ± 0.002 a	0.025 ± 0.001 ab

Letters a, b indicate significant differences, of each parameter, among all treatments (Ca(NO_3_)_2_ and CaCl_2_) using the single factor ANOVA test (*p* ≤ 0.05). Foliar spray was carried out with Ca(NO_3_)_2_ (0.5, 2, 4 kg.ha^−1^) or CaCl_2_ (3, 6 kg.ha^−1^). Control was not sprayed with Ca(NO_3_)_2_ or CaCl_2_.

**Table 3 plants-10-00245-t003:** Average of Ca contents ± S.E. (*n* = 4) in 5 transverse sections of the equatorial region (ranging from the epidermis to the center, 1–5), of tubers of *Solanum tuberosum* L., varieties Agria, Picasso and Rossi, obtained at harvest (control and treated plants with 4 kg.ha^−1^ Ca(NO_3_)_2_ and 6 kg.ha^−1^ CaCl_2_). Control was not sprayed with Ca(NO_3_)_2_ or CaCl_2_.

Ca Contents (%)					
Treatments	Agria				
	Transverse Sections in the Equatorial Region				
	1	2	3	4	5
**Control**	0.31 ± 0.02	0.05 ± 0.00	0.07 ± 0.00	0.11 ± 0.01	0.09 ± 0.00
**Ca(NO_3_)_2_ (4 kg ha^−1^)**	0.82 ± 0.04	0.33 ± 0.02	0.26 ± 0.01	0.14 ± 0.01	0.16 ± 0.01
**CaCl_2_ (6 kg ha^−1^)**	0.46 ± 0.02	0.07 ± 0.00	0.11 ± 0.01	0.16 ± 0.01	0.36 ± 0.01
**Treatments**	**Picasso**				
	**Transverse Sections in the Equatorial Region**				
	**1**	**2**	**3**	**4**	**5**
**Control**	1.03 ± 0.05	0.31 ± 0.02	0.07 ± 0.00	0.13 ± 0.01	0.47 ± 0.02
**Ca(NO_3_)_2_ (4 kg ha^−1^)**	1.02 ± 0.02	0.75 ± 0.04	1.02 ± 0.05	0.57 ± 0.03	0.66 ± 0.03
**CaCl_2_ (6 kg ha^−1^)**	1.00 ± 0.05	0.64 ± 0.03	0.34 ± 0.02	0.21 ± 0.01	0.33 ± 0.02
**Treatments**	**Rossi**				
	**Transverse Sections in the Equatorial Region**				
	**1**	**2**	**3**	**4**	**5**
**Control**	0.34 ± 0.02	0.13 ± 0.01	0.17 ± 0.01	0.21 ± 0.01	0.15 ± 0.01
**Ca(NO_3_)_2_ (4 kg ha^−1^)**	0.90 ± 0.04	0.12 ± 0.01	0.13 ± 0.01	0.13 ± 0.01	0.22 ± 0.01
**CaCl_2_ (6 kg ha^−1^)**	0.15 ± 0.01	0.13 ± 0.01	0.11 ± 0.01	0.11 ± 0.01	0.24 ± 0.02

**Table 4 plants-10-00245-t004:** Mean values ± S.E. (*n* = 4) of dry weight and total soluble solids in tubers of *Solanum tuberosum* L., varieties Agria, Picasso and Rossi, at harvest.

Treatments		Agria	Picasso	Rossi
Dry weight (%)				
**Control**		25.2 ± 0.459 a	20.9 ± 0.596 ab	26.1 ± 1.180 a
**Ca(NO_3_)_2_**	**0.5 kg ha^−1^**	21.0 ± 0.359 b	17.6 ± 0.403 c	18.1 ± 0.690 c
	**2 kg ha^−1^**	20.0 ± 0.248 b	20.9 ± 0.533 ab	21.6 ± 0.422 bc
	**4 kg ha^−1^**	20.2 ± 0.660 b	21.6 ± 0.403 ab	22.8 ± 0.410 ab
**CaCl_2_**	**3 kg ha^−1^**	22.6 ± 0.386 ab	19.7 ± 0.158 bc	23.2 ± 0.763 ab
	**6 kg ha^−1^**	22.4 ± 0.790 ab	23.0 ± 0.713 a	23.2 ± 0.090 ab
**Total Soluble Solids (°Brix)**				
**Control**		6.17 ± 0.136 a	4.83 ± 0.136 b	7.50 ± 0.408 a
**Ca(NO_3_)_2_**	**0.5 kg ha^−1^**	5.33 ± 0.272 ab	4.93 ± 0.054 b	6.17 ± 0.136 bc
	**2 kg ha^−1^**	5.40 ± 0.249 ab	5.17 ± 0.136 b	5.17 ± 0.136 c
	**4 kg ha^−1^**	5.00 ± 0.000 b	5.50 ± 0.235 ab	6.67 ± 0.134 ab
**CaCl_2_**	**3 kg ha^−1^**	5.07 ± 0.054 ab	6.51 ± 0.232 a	7.33 ± 0.272 ab
	**6 kg ha^−1^**	5.30 ± 0.244 ab	5.50 ± 0.288 ab	7.00 ± 0.288 ab

Letters a, b, c indicate significant differences, of each parameter, among all treatments (Ca(NO_3_)_2_ and CaCl_2_) using the single factor ANOVA test (*p* ≤ 0.05). Foliar spray was carried out with Ca(NO_3_)_2_ (0.5, 2, 4 kg.ha^−1^) or CaCl_2_ (3 and 6 kg.ha^−1^). Control was not sprayed with Ca(NO_3_)_2_ or CaCl_2_.

## Data Availability

Not applicable.

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
