# Peer review of "Can Foliar Pulverization with CaCl2 and Ca(NO3)2 Trigger Ca Enrichment in Solanum tuberosum L. Tubers?"

_plants, 2021, doi:10.3390/plants10020245_

Round 1
Reviewer 1 Report
The submitted by Coelho et al. manuscript contains results of a comprehensive study on the possibility of foliar calcium enrichment in Solanum tuberosum tubers. At nutritional level the intracellular Ca plays an important role in the tuberization process. In this respect, the problem addressed in the proposed manuscript is in the scope of the journal Plants. The experiments are well planned and organized. A set of up-to-date biochemical and biophysical techniques are used. The authors use proper illustrative material and statistical treatment to support their conclusions.
My notes are as follows:
- It is not clear for me from the Materials and Methods how is the spraying performed – CaCl2 and Ca(NO3)2 solutions or powder are used?
- Again, in Materials and Methods - the temperature range during photosynthetic measurements should be mentioned.
- Some abbreviations (PEI, NDVI etc.) should be explained.
- The characters in some figures (Fig.1 and Fig.4) are small and difficult to read.
- 4: There are no color circles in the legend. The fitted curves are not proper. My opinion is that it would be better to connect points with straight lines or to use only symbols.
- Some other minor corrections are included in the attached manuscript file.

Author Response
Reply to the reviewer
After reading and considering the reviewer perspectives of the paper “Can foliar pulverization with CaCl2 and Ca(NO3)2 trigger Ca enrichment in Solanum tuberosum L. tubers?”, the authors reply as follows:
Sentence of the reviewer: “The experiments are well planned and organized. A set of up-to-date biochemical and biophysical techniques are used. The authors use proper illustrative material and statistical treatment to support their conclusions.”
Reply of the authors: The authors thanks the nice sentences of the reviewer.
Sentence of the reviewer: “It is not clear for me from the Materials and Methods how is the spraying performed – CaCl2 and Ca(NO3)2 solutions or powder are used?”
Reply of the authors: The spraying of CaCl2 and Ca(NO3)2 was performed with solutions for both chemical compounds. The authors introduced this information in the Abstract and in Materials and Methods.
Sentence of the reviewer: “Again, in Materials and Methods - the temperature range during photosynthetic measurements should be mentioned.”
Reply of the authors: In Materials and Methods, the authors added a detailed information regarding the temperature range during the photosynthetic measurements (please see lines 404-413).
Sentence of the reviewer:” Some abbreviations (PEI, NDVI etc.) should be explained.”
Reply of the authors: The authors agree with the reviewer and we added an explanation of the abbreviations in the paper. Regarding the PEI and NDVI, they stand for Prince Edward Island (Nutrient Management) and Normalized Difference Vegetation Index, respectively (please see lines 93 and 123, respectively)
Sentence of the reviewer: “The characters in some figures (Fig.1 and Fig.4) are small and difficult to read”
Reply of the authors: The authors change the characters of the figures (Fig.1 and Fig. 4) for better reading.
Sentence of the reviewer:” Fig.4: There are no color circles in the legend. The fitted curves are not proper. My opinion is that it would be better to connect points with straight lines or to use only symbols.”
Reply of the authors: In Fig. 4 their color circles in the legend were added. We understand the perspective of the reviewer, however, the authors agree that is better to connect the points with a polynomial trend line rather than straight lines.
Sentence of the reviewer: “Some other minor corrections are included in the attached manuscript file”
Reply of the authors: The authors thank the reviewer and have made the minor corrections marked by the reviewer.
Reviewer 2 Report
Comments to authors
General comment: Authors used different modern technologies and methods to measure impact of foliar Ca application by spraying. My major concern is the presentation of the results. Namely, authors set experiment in three fields but results are presented as one (comparison of control and treatment per cultivar) regardless of the experimental field. Why is that? In Tables n =4, what that means? Four tubers analyzed from all 3 fields? Technical replicate, biological replicate? It is not clear. Data on yield is missing as well as description of measured parameters such as colorimetric analyses of what?
Abstract
You mentioned changes in the yield and total soluble sugars, I don’t see any of these data in the results?
Introduction
Lines 63-64 Break in two sentences.
Lines 79-80 The sentence is not clear, please rephrase.
Lines 80, 83… You cannot start sentence with the reference number, Write the name of the first author. Please correct this thorough the whole manuscript.
Line 92 Define abbreviation PEI
Line 94 …”35 found that...” write author name. Correct similar sentences in the whole manuscript.
Line 100 correct to species
Results:
Lines 114-115 Rephrase the sentence.
It is not clear why authors present their results as one result regardless of the experimental field?
Table 2. In which field? All 3?
Figure 2. correct Kg into kg in axis legend
Lines 97-198 total soluble solids, please provide explanation on what they include (which parameters?).
Lines 207-209 Please explain these results. What these specters mean and represent? Any particular changes physiological characteristics due to Ca treatment?
M&M section:
1) Did you use any kind of basal fertilization during field trails? If yes, please provide the details. Why did you choose these cultivars over the others? Describe their food/industrial value.
2) Line 364 : What does these numbers in brackets represents? Are they catalogue numbers of kits? If they are, please rephrase the sentence using only manufacturer’s name.
3) Lines 367-368 Please rephrase the sentence… , at 20℃ reference temperature to determine…
4) How many plots were in each experimental field? Four? Please add description in lines 374-375 ? What represents biological and technical replicates? I don’t understand, from each plot you measured 4 tubers? How many tubers form each field you measured? Describe the size of the tubers used for measurements? Where they approximately same size?
5) Lines 405-406 check grammar in the sentence
Conclusion
I don’t understand the last sentence, particular “… foliar spraying kept.”? Is it missing something?
Please provide information on agronomical implications of your research regarding cultivars used in this study.
Author Response
Reply to the reviewer
After reading and considering the reviewer perspectives of the paper “Can foliar pulverization with CaCl2 and Ca(NO3)2 trigger Ca enrichment in Solanum tuberosum L. tubers?”, the authors reply as follows:
Sentence of the reviewer:” My major concern is the presentation of the results. Namely, authors set experiment in three fields but results are presented as one (comparison of control and treatment per cultivar) regardless of the experimental field. Why is that? In Tables n =4, what that means? Four tubers analyzed from all 3 fields? Technical replicate, biological replicate? It is not clear. Data on yield is missing as well as description of measured parameters such as colorimetric analyses of what?”
Reply of the authors: Concerning to the usage of three experimental field, we consider all of them as a unique test system, as our main goal was to assess the efficiency of Ca enrichment in tubers of three species, through foliar spraying with CaCl2 and Ca(NO3)2, but also implicating soil heterogeneity of three potato-growing fields.
In tables, n=4 means that 4 tubers were analyzed per treatment in each field.
Regarding the yield, it was a mistake to mention it in the abstract. In fact, considering the main aims of the paper, after discussion among all the authors, we considered that these data was not relevant for the conclusions and, therefore, during the rewriting, we forgot to remove it from the abstract.
Concerning the description of colorimetric analyses are explain in lines 447-453.
Sentence of the reviewer: ”You mentioned changes in the yield and total soluble sugars, I don’t see any of these data in the results?”
Reply of the authors: Concerning to the yield, please see the last answer. Relatively to total soluble solids, please see Table 4. There some significant differences can be observed in each variety.
Sentence of the reviewer: “Lines 63-64 Break in two sentences.
Lines 79-80 The sentence is not clear, please rephrase
Lines 80, 83 You cannot start sentence with the reference number, Write the name of the first author. Please correct this thorough the whole manuscript.
Line 94 …”35 found that...” write author name. Correct similar sentences in the whole manuscript.
Line 100 correct to species”
Reply of the authors: The authors made the changes suggested by the reviewer.
Sentence of the reviewer: “Define abbreviation PEI”
Reply of the authors: The authors already explain the abbreviation to Reviewer 1 (please see line 93)
Sentence of the reviewer: “Lines 114-115 Rephrase the sentence.
Reply of the authors: The authors made the change proposed by the reviewer Please see lines 121-123)
Sentence of the reviewer: “It is not clear why authors present their results as one result regardless of the experimental field? Table 2. In which field? All 3? Figure 2. correct Kg into kg in axis legend”.
Reply of the authors:
Table 2 shows the mean values of Ca in tubers regarding the different treatments of the three varieties. As reported in Materials and Methods, in each field was cultivated one variety. Agria variety was cultivated in Moledo field, Picasso variety in Casal Galharda field, and Rossi variety in Boas Águas field.
Regarding the Fig.2 axis legend, the authors made the change to kg.
Sentence of the reviewer: “Lines 97-198 total soluble solids, please provide explanation on what they include (which parameters?).”
Reply of the authors: The total soluble solids mostly include sugars – sucrose, glucose, and fructose (please see lines 444-445).
Sentence of the reviewer: “Lines 207-209 Please explain these results. What these specters mean and represent? Any particular changes physiological characteristics due to Ca treatment?”
Reply of the authors: Being color, one of the quality parameters in food, the results show that there wasn’t any change in the pulp color of the tubers regarding the Ca applications (please see lines 218-223 and 447-453). In this context, means that the quality of the biofortified tubers are similar to the control ones.
Sentence of the reviewer: “Did you use any kind of basal fertilization during field trails? If yes, please provide the details. Why did you choose these cultivars over the others? Describe their food/industrial value.”
Reply of the authors: Concerning to the basal fertilization, the usual set of actions for potato cultivation was introduced. Briefly, this procedure was introduced in the paper (please see lines 359-366).
The varieties Agria, Picasso and Rossi particularly relevant potato varieties at a commercial / industrial levels. A short explanation was introduced in the end of the introduction (please see lines 106-110).
Sentence of the reviewer: “Line 364. What does these numbers in brackets represents? Are they catalogue numbers of kits? If they are, please rephrase the sentence using only manufacturer’s name.”
Reply of the authors: The numbers in brackets represents the catalogue numbers of kits from Sectroquant® from Merck KGaA. The sentence was rephrased by the authors (Please see lines 386 - 388).
Sentence of the reviewer: “Lines 367-368 Please rephrase the sentence… , at 20℃ reference temperature to determine… “
Reply of the authors: The authors rephrased the sentence (please see lines 391-393).
Sentence of the reviewer: “How many plots were in each experimental field? Four? Please add description in lines 374-375 ? What represents biological and technical replicates? I don’t understand, from each plot you measured 4 tubers? How many tubers form each field you measured? Describe the size of the tubers used for measurements? Where they approximately same size?”
Reply of the authors: Each plot contained the six treatments (control included), having been carried out in quadruplicate (please see lines 397-399). Concerning to replicates, for each analysis tubers were collected from all replicates, with similar sizes (please see lines 423- 450)
Sentence of the reviewer: ”Lines 405-406 check grammar in the sentence”
Reply of the authors: The authors rephrased the sentence.
Sentence of the reviewer:” I don’t understand the last sentence, particular “… foliar spraying kept.”? Is it missing something?
Please provide information on agronomical implications of your research regarding cultivars used in this study.”
Reply of the authors: The authors rephrased and complemented the last sentence of the paper (please see lines 465- 469).
Round 2
Reviewer 2 Report
Comments to authors
In Abstract you wrote again about changes in the total soluble sugars but in the results and discussion you talk about total soluble solids. You have to be consistent with the terms you use. Use either total soluble sugars or total soluble solids in the manuscript. Because there are methods that measure only total soluble sugars, such as Anthrone assay, in different plant species.
Line 83 reference 29 it is not correctly cited in the text.
Line 96 you did not corrected reference …” [35] found that…“ Please write authors names
Line 98 you did not corrected reference “…potato tubers and [37], spraying with…“ Please write authors names
Line 238 check grammar in the sentence
Line 430 check grammar in the sentence
Just to be clear, from each plot (4 plots per each cultivar in one experimental field) you selected 2 tubers (total of 8) for Ca measurement (line 410) and for dry weight you selected one tuber per each plot (lines 429 and 430)? So, 4 plots represent biological replicates (n=4) as you mentioned in Tables, and 8 tubers for e.g. Ca measurement or 1 tuber for dry weight represent technical replicate?
Did you consider measuring more than one tuber per plot for dry weight and colorimetric analysis? Do you think is enough to measure one tuber per plot to draw significant conclusions (total of 4 tubers)? Explain why?
I am asking these questions because you did not answer it in previous revision (What represents biological and what represents technical replicate?) This should be clearly stated.
Author Response
Paper submitted to Plants, with nº1064078
The authors of the manuscript answer to reviewer 2 as follows:
Comment of the reviewer: “Abstract you wrote again about changes in the total soluble sugars but in the results and discussion you talk about total soluble solids. You have to be consistent with the terms you use. Use either total soluble sugars or total soluble solids in the manuscript. Because there are methods that measure only total soluble sugars, such as Anthrone assay, in different plant species.”
Reply of the authors: In the abstract, as suggested, “total soluble sugars” were replaced by “total soluble solids”.
Comment of the reviewer: “Line 83 reference 29 it is not correctly cited in the text.”
Reply of the authors: “As indicated by the reviewer, the reference was corrected, changing “Opar” by “Oparka and Davies”.
Comment of the reviewer: “Line 96 you did not corrected reference …” [35] found that…“ Please write authors names”
Reply of the authors: As indicated by the reviewer, authors names were introduced. Accordingly, now it is “… El-Hadidi et al. [35] found that…”.
Comment of the reviewer: “Line 98 you did not corrected reference “…potato tubers and [37], spraying with…“ Please write authors names”
Reply of the authors: As indicated by the reviewer, authors names were introduced. Accordingly, now it is “…potato tubers and Seifu, and Deneke [37], spraying with…“.
Comment of the reviewer: “Line 238 check grammar in the sentence“
Reply of the authors: Due to the track change system, it was not clear for the authors, which sentence the reviewer considered. Nevertheless, we believe that it was the one, which we are pointing bellow. Thus, as suggested by the reviewer, the sentence was changed. Accordingly, now it is “In this context, the experimental areas in each potato fields were selected (Figure 1) to minimize the effects of runoff water [38].”.
Comment of the reviewer: “Line 430 check grammar in the sentence“
Reply of the authors: Due to the track change system, it was not clear for the authors, which sentence the reviewer considered. Nevertheless, we believe that it was the one, which we are pointing bellow. Thus, as suggested by the reviewer, the sentence was changed. Accordingly, now it is “Total soluble solids (which are mostly an indication sucrose, glucose, and fructose) was also measure in the juice of four randomized tubers (one of each replicate) per treatment, using a digital refractometer Atago (Atago, Tokyo, Japan)”.
Comment of the reviewer: “Just to be clear, from each plot (4 plots per each cultivar in one experimental field) you selected 2 tubers (total of 8) for Ca measurement (line 410), and for the dry weight you selected one tuber per each plot (lines 429 and 430)? So, 4 plots represent biological replicates (n=4) as you mentioned in Tables, and 8 tubers for e.g. Ca measurement or 1 tuber for dry weight represent technical replicate?”
Did you consider measuring more than one tuber per plot for dry weight and colorimetric analysis? Do you think is enough to measure one tuber per plot to draw significant conclusions (total of 4 tubers)? Explain why?
Reply of the authors: We think that there the reviewer confused plots with replicates. For us, one plot it is an area where the field assay was carried out, whereas a replicate considered samples collected inside the plot. Therefore, each variety had one plot (with different treatments) and within each treatment, a specific number of potatoes were collected, joined, and a final extract was prepared for the analytical assay, being four samples read from each final extract (i.e., four replicates). For instance, for Ca measurements (through atomic absorption), 8 tubers were collected for each treatment and after the final extract preparation, 4 readings were carried out (n=4). In the case of the dry weight measurements, total soluble solids, and colorimetric, 4 tubers were collected from each treatment (i.e., 4 replicates, thus n=4). Nevertheless, in the materials and methods section, we changed several sentences just to improve the understanding of the procedures.
Accordingly, the sentences “After harvest, considering the four replications in the experimental potato-producing fields, eight randomized tubers (with similar size) were washed, dried at 60 ºC until constant weight, and grounded in an agate mortar. After that, acid digestion procedure was performed with a mixture of HNO3- HClO4 (4:1) according to [63,64]. After filtration, Ca content was measured by atomic absorption spectrophotometry, using a model Perkin Elmer AAnalyst 200, and the absorbency in mg / L was determined with a coupled AA WinLab software.” were changed to “After harvest in the experimental potato-producing fields, eight randomized tubers (with similar size) were washed, dried at 60 ºC until constant weight and grounded in an agate mortar. After that, the homogenate was divided in four samples (n=4) and an acid digestion procedure was performed with a mixture of HNO3- HClO4 (4:1) according to [63,64]. After filtration, Ca content was measured by atomic absorption spectrophotometry, using a model Perkin Elmer AAnalyst 200, and the absorbency in mg / L was determined with a coupled AA WinLab software.
Additionally, the sentences “Dry weight was performed considering four randomized tubers (one of each replicate) per treatment. Total soluble solids (which are mostly an indication sucrose, glucose, and fructose) was also measure in the juice of four randomized tubers (one of each replicate) per treatment, using a digital refractometer Atago (Atago, Tokyo, Japan). Colorimetric parameters were determined in the pulp of four fresh tubers (one of each replicate) per treatment with a scanning spectrophotometric colorimeter (Agrosta, European Union).” were changed to “Dry weight was performed considering four randomized tubers per treatment (n=4). Total soluble solids (which are mostly an indication sucrose, glucose, and fructose) was also measure in the juice of four randomized tubers per treatment (n=4), using a digital refractometer Atago (Atago, Tokyo, Japan). Colorimetric parameters were determined in the pulp of four fresh tubers per treatment (n=4) with a scanning spectrophotometric colorimeter (Agrosta, European Union).”
Comment of the reviewer: “I am asking these questions because you did not answer it in previous revision (What represents biological and what represents technical replicate?) This should be clearly stated.”
Reply of the authors: The authors thanks the reviewer for these kind reminders. In fact, differences between biological and technical readings were not clearly stated. As it can be seen in the previous reply to the reviewer, these aspects were now objectively defined in the Materials and Methods section. Measurement of Ca considered the mean value of 8 tubers of each treatment inside the plot, considering 4 samples individually prepared after homogenization, whereas the other analysis considered the mean value of 4 individual tubers of each treatment, measured individually.
15 January 2021
By the authors
Round 3
Reviewer 2 Report
I read the revised version of the manuscript. Authors improved all the sections of the manuscript as I requested. I only have some minor corrections:
Lines 450-451 in the sentence „Total soluble solids (which are mostly an indication sucrose, glucose, and fructose…“
Correct the sentence to “…. an indication of sucrose, glucose….”
Author Response
Paper submitted to Plants, with nº1064078
The authors of the manuscript answer to reviewer 2 as follows:
Comment of the reviewer: “I read the revised version of the manuscript. Authors improved all the sections of the manuscript as I requested. I only have some minor corrections:
Lines 450-451 in the sentence “Total soluble solids (which are mostly an indication sucrose, glucose, and fructose…“
Correct the sentence to “…. an indication of sucrose, glucose….” “
Reply of the authors: As indicated by the reviewer, the sentence “Total soluble solids (which are mostly an indication sucrose, glucose, and fructose…“ was replaced to “ “Total soluble solids (an indication of sucrose, glucose, and fructose…“.